# The Effect of Electrical Polarity on the Diameter of Biobased Polybutylene Succinate Fibers during Melt Electrospinning

**DOI:** 10.3390/polym14142865

**Published:** 2022-07-14

**Authors:** Maike-Elisa Ostheller, Naveen Kumar Balakrishnan, Robert Groten, Gunnar Seide

**Affiliations:** 1Aachen-Maastricht Institute for Biobased Materials (AMIBM), Maastricht University, Brightlands Chemelot Campus, Urmonderbaan 22, 6167 RD Geleen, The Netherlands; gunnar.seide@maastrichtuniversity.nl; 2Aachen-Maastricht Institute for Biobased Materials e.V. (AMIBM e.V.), Lutherweg 2, 52068 Aachen, Germany; naveen.balakrishnan@maastrichtuniversity.nl; 3Department of Textile and Clothing Technology, Niederrhein University of Applied Sciences, Campus Moenchengladbach, Webschulstrasse 31, 41065 Moenchengladbach, Germany; robert.groten@hs-niederrhein.de

**Keywords:** melt electrospinning, nonwoven, machine optimization, electrical field polarity

## Abstract

Melt electrospinning is a simple, versatile, and widely used technique for the production of microfibers and sub-microfibers. Polybutylene succinate (PBS) is a promising raw material for the preparation of melt-electrospun fibers at the laboratory scale. The inclusion of additives in the PBS melt can reduce the final fiber diameter, but economically feasible larger-scale processes remain challenging. The fiber diameter can also be reduced by machine optimization, although this is expensive due to the complexity of melt-electrospinning devices. Changes in electrical field polarity have provided a low-cost strategy to reduce the diameter of fibers produced by solution-electrospinning, but there is little information about the effect of this parameter on the final diameter of melt-electrospun fibers. We therefore determined the effect of field polarity on the diameter of melt-electrospun PBS fibers at the laboratory scale and investigated the transferability of these results to our 600-nozzle pilot-scale device. Changing the polarity achieved a significant reduction in fiber diameter of ~50% at the laboratory scale and ~30% at the pilot scale, resulting in a minimum average fiber diameter of 10.88 µm. Although the effect of field polarity on fiber diameter was similar at both scales, the fibers in the web stuck together at the laboratory scale but not at the pilot scale. We have developed an inexpensive method to reduce the diameter of melt-electrospun PBS fibers and our data provide insight into the transferability of melt electrospinning from the laboratory to a pilot-scale machine.

## 1. Introduction

Electrospinning is a simple and cost-effective process for the manufacturing of microfibers and nanofibers with many advantages over alternative techniques, such as drawing, template synthesis, and phase separation [1]. Conventional fiber-spinning processes, such as melt spinning and wet spinning, draw and stretch fibers by applying a mechanical force, whereas electrospinning uses an electric field to produce continuous fibers. The typical electrospinning setup includes a syringe filled with polymer solution (solution-electrospinning) or polymer melt (melt electrospinning), a collector plate to deposit the resulting fiber, and a high-voltage supply to provide the electric field by establishing a potential difference between the end of the needle capillary and the collector [2]. Fibers are produced when the electrostatic repulsive force of the polymer solution or melt overcomes the surface tension, resulting in the formation of a so-called Taylor cone, which emits an electrically charged polymer jet. The charge density of the jet interacts with the external field and causes whip-like movements that stretch the fiber and reduce its diameter to the micrometer or nanometer range. The continuous filaments are deposited as a nonwoven-like material on the collector [3].

Polymer microfibers and nanofibers have several valuable characteristics, including a very large surface-area-to-volume ratio, flexibility in surface functionalities, and superior mechanical properties, such as stiffness and tensile strength [4]. Such fibers are therefore promising for many different applications, including filters, porous materials [5,6], membranes, tissue engineering scaffolds, wound dressings, and sound absorption matrices [7,8]. The lower viscosity and higher electrical conductivity of polymer solutions means that solution-electrospinning currently produces thinner fibers than melt electrospinning and remains the favored process. However, only 2–10% of the liquid processed during solution-electrospinning is the polymer, with the remainder being the toxic solvent that evaporates during processing [2]. The high evaporation rate of the solvent, the strong dependency on the elasticity of the polymer solution, and the low flow rates needed to produce narrow fibers all reduce the productivity of solution-electrospinning and thus hinder its industrial application [9].

Melt electrospinning is a more efficient process and the absence of solvents makes it more environmentally sustainable, but the minimum fiber diameter achieved at the laboratory scale is in the low micrometer range [10,11]. Several challenges must be overcome to achieve the larger-scale production of nanofibers [12,13,14]. One of these challenges is the high viscosity and low electrical conductivity of polymer melts [13,15,16,17], which can be addressed in part by the inclusion of additives [18]. For example, in our previous study, we investigated the use of curcumin and silver nanoparticles as multifunctional additives to reduce the viscosity of molten polybutylene succinate (PBS) while increasing its electrical conductivity [11]. This reduced the fiber diameter compared to pure PBS but did not come close to the nanoscale fiber diameters typically achieved by solution-electrospinning.

The fiber diameter can also be reduced by modifying the melt-electrospinning device. For example, gas-assisted melt-electrospinning (GAME) reduced the diameter of melt-electrospun PLA fibers by 10% [19], and a multi-temperature control unit also reduced the diameter of melt-electrospun PLA fibers [20]. Other approaches have been discussed in a recent review [21]. However, all of these approaches require expensive machine add-ons or upgrades and have so far been exclusively studied on lab-scale devices, including a single nozzle. Today, studies about the transferability of machine modifications to pilot scale are missing. A much less costly approach is the adjustment of machine parameters such as the electrical field strength and its polarity. Several configurations of high voltage and ground connections can influence the electric field strength and thus the properties of the resulting fibers [21,22,23], and the three most common setups are shown in Figure 1. 

The conventional solution-electrospinning setup features a syringe with a static (usually positive) charge, while the collector is grounded (Figure 1a) [24]. Several groups have investigated the effect of reversing this configuration by applying a high (positive) voltage to the collector while grounding the syringe (Figure 1b), but this generates relatively small coulombic repulsion forces on the jet during solution-electrospinning, and the conventional setup is therefore more efficient, producing narrower nanofibers with finer and more homogenously distributed pores [21,24]. A solution-electrospinning setup with a direct positive charge applied to the nozzle and an indirect negative charge on the (grounded) collector generated a larger electric current and a stronger electric field force applied to the jet, resulting in an unstable region in the flying trajectory that reduced the fiber diameter [21]. Applying a charge to the syringe while grounding the collector remains the most common solution-electrospinning setup.

In a few melt-electrospinning studies, a charge was applied to both the nozzle and the collector [22], but in most cases the collector has a positive charge and the nozzle is grounded (Figure 1c) [23]. Directly charging the polymer melt may promote arcing from the charged spinneret to other conductive metallic components in the polymer delivery system (extruder), destroying the equipment [24]. It therefore remains challenging to apply a direct positive charge to the spinneret. To avoid the risk of damage, a negative charge can be applied to the collector. Although the modification of electrical polarity has led to improvements in solution-electrospinning, the effect of electrical polarity on melt-electrospun fiber has not been explored in detail [23]. We therefore determined the effect of changing the polarity of the collector during the melt-electrospinning of PBS in order to minimize the final fiber diameter. We also investigated the transferability of these results to our 600-nozzle pilot-scale device. Our results can be used to tailor the melt electrospinning of biobased polymers for the production of narrower fibers.

## 2. Materials and Methods

### 2.1. Materials

For all experiments we used biobased PBS fiber-grade resin (FZ78TM) produced by the polymerization of biobased succinic acid and 1,4-butanediol. The manufacturer (MCPP, Düsseldorf, Germany) reported the following specifications: melt flow rate = 22 g/10 min at 190 °C using a weight of 2.16 kg and a crystalline melting temperature of 115 °C. The polymer was vacuum dried at 60 °C for 12 h before processing.

### 2.2. Melt-Electrospinning Equipment

Our laboratory-scale single-fiber melt-electrospinning device (Figure 2a) was used as a starting point for optimization. This device includes a JCS-33A temperature process controller (Shinko Technos, Osaka, Japan) and a PT 100 platinum thermocouple (Omega Engineering, Deckenpfron, Germany) to control the melting temperature, a KNH65 high-voltage power supply (Eltex-Elektrostatik, Weil am Rhein, Germany) with a ±6–30 kV range (positive voltage on the collector and a grounded nozzle tip), heating elements, an 11 Plus spin pump (Harvard Apparatus, Cambridge, MA, USA) with a feed rate of 0.1–4.0 mL/min, a 2 mL glass syringe (Poulten & Graf, Wertheim, Germany) equipped with an additional metal orifice of 0.90 mm serving as the spinneret nozzle, and a flat aluminum plate (6 cm diameter) overlaid with thin paperboard serving as the collector. Before the experiments, we replaced the original heating chamber, consisting of several individual parts, with a continuous heating tube tailored for the syringe and a thread to attach the metal nozzle (which was grounded), allowing better control of heat exchange (Figure 2b). Trials were carried out at a polymer melt temperature of 235 °C. The process parameters are summarized in Table 1.

To evaluate the transferability of the process, we used a newly developed prototype pilot-scale setup including a spinneret with 600 nozzles, each 0.3 mm in diameter and spaced at 8 mm intervals [1] (Figure 3b). The pilot-scale machine uses a more efficient heating system, a revised nozzle to reduce dwell times, and an integrated collector design and fiber deposition concept [25,26]. A speed-adjustable single-screw extruder with three heating zones based on integrated heating elements and a spinning pump ensures a constant supply of polymer melt. Heating elements around the spinneret are used to avoid the blocking of individual nozzles caused by rapid polymer solidification. Melt-electrospinning was carried out at extruder and spinneret temperatures of 235 °C. A 34 × 14 cm aluminum plate collector allows the production of continuous nonwovens up to 340 mm in width. The process parameters are summarized in Table 2. Laboratory-scale and pilot-scale melt electrospinning have been both carried out on the same day, at a room temperature of 22.7 °C and a relative humidity of 63%.

### 2.3. Characterization of PBS Fibers

The electrical resistance between the nozzle and ground was determined using a STIER Profi Multimeter (Stier Industrial, Berlin, Germany). Fiber diameters were measured using an Olympus BX53 microscope fitted with an Olympus DP26 camera (Olympus, Leiderdorp, Netherlands). The fiber diameter was measured 100 times for each sample in different positions, based on the 50× magnified image.

We also measured degradation in terms of molecular weight (after lab scale and pilot scale melt-electrospinning) by gel permeation chromatography (GPC) using a 1260 Infinity System (Agilent Technologies, Santa Clara, CA, USA). Hexafluor-2-isopropanol (HFIP) containing 0.19% sodium trifluoroacetate was used as the mobile phase at a flow rate of 0.33 mL/min. Solutions were prepared by dissolving 5 mg of PBS fibers in HFIP for ~2 h, passing the solutions through a 0.2 µm polytetrafluoroethylene filter and injecting them into a modified silica column filled with 7 µm particles (Polymer Standards Service, Mainz, Germany). The experiment was calibrated against a standard polymethyl methacrylate polymer (1.0 × 10^5^ g/mol), and the relative weight average molecular weight (Mw), number average molar mass (Mn), and polydispersity index (PDI) of each fiber were recorded and compared.

## 3. Results and Discussion

### 3.1. Effect of Machine Modification on the Laboratory-Scale Melt-Electrospinning Process

We optimized our laboratory-scale melt-electrospinning setup by replacing the original heating element with a continuous heating tube to better control heat exchange with the polymer. In our previous setup, it was necessary to set the heating element to 250 °C to achieve a polymer melt temperature of 235 °C, which inhibited processing because longer exposure times caused the polymer nearest the wall of the syringe to degrade. By modifying the heating element, we achieved a polymer melt temperature of 235 °C by setting the heating element to the same temperature, reducing the likelihood of polymer degradation. The heat in the polymer can also build up due to shear in the extruder, and because that is missing in a glass syringe system, temperature control is more challenging and critical.

In addition to better heat exchange, the electrical resistance measured at the tip of the nozzle fell from 200 MΩ to 20 Ω. This was also anticipated because the original heating system featured non-continuous connections between individual elements. The resulting air gaps therefore caused inconsistent grounding in the heating system, increasing the electrical resistance at the nozzle tip. This was overcome by grounding the new continuous heating tube. Furthermore, the original setup did not induce whipping behavior below an applied potential difference of 50 kV, whereas the optimized equipment induced whipping behavior at 30 kV. The low resistance at the nozzle tip creates a more concentrated electrical field that favors the electrospinning process. We therefore used the optimized setup for all further laboratory-scale tests (Figure 2b).

### 3.2. Effect of Polarity on Fiber Diameter and Distribution in the Laboratory-Scale Process

Figure 4 shows the average fiber diameter (±standard deviation) achieved using our optimized laboratory-scale setup at a production rate of 0.1 mL/min, a polymer melt processing temperature of 235 °C, and an electric field strength of +30 kV (left) and −30 kV (right). In both cases, we tested nozzle diameters of 0.1, 0.3, and 0.6 mm as well as nozzle-to-collector distances of 6, 8, and 10 cm. We compared tests carried out under identical conditions but opposing polarities to isolate the effect of polarity on fiber diameter.

In agreement with our previous reports, the larger nozzle diameter (0.6 mm) significantly increased the fiber diameter at all nozzle-to-collector distances at an applied potential difference of +30 kV [10]. Because the polymer melt emerging from the nozzle is pulled down by gravity and by its interaction with the electric field, the polymer jet is likely to be pushed out more quickly from the wider nozzle, and the greater mass of polymer has less time to interact with the electric field, thus reducing the probability of whipping behavior and increasing the fiber diameter. Additionally, consistent with our previous reports, increasing the nozzle-to-collector distance had a less significant effect on the fiber diameter when using larger nozzles. Accordingly, the average diameter of fibers produced by the 0.6-mm nozzle at a distance of 8 cm from the collector was 82.92 µm. This is ~40% narrower than the fibers produced at a distance of 10 cm (115.53 µm), suggesting that increasing the distance reduces the electric field strength and thus the interaction between the polymer jet and electric field. However, a nozzle-to-collector distance of 6 cm did not reduce the fiber diameter any further—indeed the average diameter increased to 99.41 µm. This distance may give the polymer jet insufficient time to interact with the electric field before deposition, thus resulting in thicker fibers. Contrary to our expectations, increasing the nozzle-to-collector distance for nozzles of 0.1 and 0.3 mm diameter did not reduce the fiber diameter. With the 0.1-mm nozzle, nozzle-to-collector distances of 10, 8, and 6 cm resulted in fibers with average diameters of 66.99, 71.12, and 70.81 µm, respectively. With the 0.3-mm nozzle, nozzle-to-collector distances of 10, 8, and 6 cm resulted in fibers with average diameters of 91.87, 69.68, and 64.26 µm, respectively. These results indicate that the best nozzle-to-collector distance should be evaluated for each nozzle size. Furthermore, we previously found that 10 cm is the optimal nozzle-to-collector distance when the field strength is 50 kV, but this does not appear to be the case at 30 kV, suggesting that the optimal nozzle-to-collector distance should be established for different field intensities too.

Although Taylor cone formation and whipping behavior were observed under all conditions tested when the electric field strength was +30 kV, whipping behavior was enhanced when the polarity was reversed, significantly reducing the average fiber diameter regardless of the other parameters. This may reflect the 1000-fold difference in mass between a proton and an electron, which would cause protons to migrate more slowly than electrons in a field of equal magnitude [27]. Protons would therefore move more slowly toward the edge of the negatively charged nozzle compared to electrons moving toward a positively charged collector. Because protons would not have sufficient time to reach the needle, the polymer jet ejected from the tip of the nozzle would also carry a few protons toward the collector (Figure 5).

With a negatively charged collector (Figure 5a), the flying jet carries a strong positive charge and moves in accordance with the electric field force and the gravitational force, undergoing extensive whipping movements that generate thinner fibers. In contrast, with a positively charged collector (Figure 5b), the jet takes only a partial negative charge before travelling to the collector plate, resulting in a lower net charge and smaller electric current and thus a more stable jet that generates coarser fibers. Furthermore, the electric field force travels in the same direction as the gravitational force when the collector is negatively charged, which may produce thinner fibers that are more uniformly deposited. When the collector is positively charged, the electric field opposes the gravitational force pulling the fibers down, which could disrupt the spinning process and cause the non-uniform deposition of fibers.

The laboratory-scale experiments showed that the most significant reduction in average fiber diameter was achieved by applying a negative charge to the collector and using a 0.1-mm nozzle. Under these conditions, the average fiber diameter was reduced by ~45% (from 70.81 to 39.40 µm) at a nozzle-to-collector distance of 6 cm, by ~55% (from 71.12 to 32.36 µm) at 8 cm, and by ~40% (from 66.99 to 40.47 µm) at 10 cm. The average fiber diameter at each distance increased when using wider nozzles, suggesting that the larger mass of the polymer is also less affected by the electric field when the polarity is reversed. With larger nozzles, the time required for negative charges to reach the wall is also slightly longer, so the polymer jet would contain more negative charges, reducing the effect of the electric field.

Increasing the distance between the 0.1-mm nozzle and the collector appeared to have no significant influence on the final fiber diameter. At a 6 cm distance, the average fiber diameter was 39.40 µm. This fell to 32.26 µm at 8 cm but increased to 40.47 µm at 10 cm. When switching to the 0.3-mm nozzle, the average fiber diameter was ~20% lower with the negatively charged collector at all distances (52.13 µm at 6 cm, 58.22 µm at 8 cm, and 75.07 µm at 10 cm). When switching to the 0.6 mm nozzle, the average fiber diameter was not affected to the same degree by the polarity, but the negatively charged collector nevertheless reduced the diameter by 6% at 6 cm distance (from 99.41 to 93.89 µm), by 17% at 8 cm (from 82.92 to 68.46 µm), and by 5% at 10 cm (from 115.53 to 105.18 µm).

The non-woven-like fiber webs produced using negatively and positively charged collectors differed in surface coverage and distribution (Figure 6). Fibers produced using the negatively charged collector covered more of the surface, appeared to be packed more tightly, and were distributed more uniformly (Figure 6a) than the fibers produced using the positively charged collector (Figure 6b). This may reflect the longer helical motion of the flying jet triggered by the negative charge on the collector, because similar observations were reported during the pilot-scale preparation of cellulose acetate and polyvinylpyrrolidone fibers by solution-electrospinning [27], suggesting that a negatively charged collector may also improve pilot-scale melt-electrospinning.

### 3.3. Effect of Polarity on Fiber Diameter and Distribution in the Pilot-Scale Process

Figure 7 shows the average fiber diameter (±standard deviation) produced at an extruder and spinneret temperature of 235 °C, a spin pump speed of 2, 5, or 10 rpm, a nozzle-to collector distance of 7, 8, or 10 cm, and potential differences +30 or −30 kV. As observed in the laboratory-scale process, changing the polarity of the collector from positive to negative reduced the diameter of the resulting PBS fibers.

Regardless of the polarity of the collector, the spin pump speed of 2 rpm produced the narrowest fibers at all nozzle-to-collector distances. The fiber diameter increased with increasing spin pump speed when using the negatively charged collector, but there was no significant difference between spin pump speeds of 5 and 10 rpm when using the positively charged collector. This may reflect the thicker polymer jet formed at 5 rpm. Given that protons move very slowly toward the edge of the negatively charged nozzles, the polymer jet ejected from the tip of each nozzle carries the maximum number of protons towards the collector and increasing the spin pump speed would not increase the number of protons any further. The nozzle-to-collector distance had no significant influence on the fiber diameter at spin pump speeds of 5 and 10 rpm, which lends support to the previous hypothesis. The lowest average fiber diameter (10.88 µm) was achieved at a spin pump speed of 2 rpm and a distance of 8 cm using the negatively charged collector.

The fiber web samples produced by running the pilot-scale equipment at 2 rpm with a nozzle-to-collector distance of 7 cm again differed in structure depending on the polarity of the electric field (Figure 8). The fibers produced using the negatively charged collector were arranged in a more uniform manner (Figure 8a), possibly due to the longer helical motion of the polymer flying jet as stated above [27]. In contrast, the fibers produced using the positively charged collector were arranged unevenly, with isolated polymer melt drops indicating that the electric field strength was probably insufficient to produce an even sample (Figure 8b).

### 3.4. Molecular Weight of the Melt-Electrospun Fibers

The GPC curves of PBS fibers from laboratory-scale and pilot-scale both processed at 235 °C are presented in Figure 9.

The relative Mw of extruded PBS fibers at laboratory-scale was 99,440 Da, the Mn was 22,560 Da, and the PDI was 4.41. Even if the dwell time is much greater for the pilot-scale device (~45 min) in comparison to the laboratory-scale machine (~2 min), the results do not indicate any degradation of PBS (Table 3). Contrary to our expectations, these values increased significantly for the fibers extruded at pilot-scale for all spin pump speeds. We therefore hypothize that the additives used during the production of PBS granules, such as initiator and catalysts, continue to polymerize due to the high dwell time during the processing of PBS at the pilot-scale device at all spin pump speeds, forcing chain extension that leads to an increase in Mn and Mw. The different spin pump speeds investigated during pilot-scale melt electrospinning do not lead to significant changes within Mn, Mw, and PDI values (Table 3). We believe that the difference in throughput at the different spin pump speeds is too low to have a significant impact on Mn and Mw.

### 3.5. Insights from the Transferability of Results from Laboratory to Pilot Scale

Although the general principle of melt-electrospinning is the same at the laboratory and pilot scales, there are fundamental differences between the systems that must be considered. The single-nozzle device generally has a much shorter dwell time (2 min). Here, the polymer melt is exposed only to the shear in the nozzle because there is no additional extruder. In addition, the single-nozzle device has a collector surface area of 24.63 cm^2^, and there is no interaction with or interference caused by polymer jets from adjacent nozzles. At the laboratory scale, switching from positive to negative polarity always reduced the fiber diameter, although the effect was much more significant with smaller nozzles. As stated above, the negatively charged collector resulted in the deposition of fibers more uniformly, more densely, and over a larger surface area and the fibers stuck to each other and thereby formed a non-woven-like structure.

The pilot-scale process has a much longer dwell time (45 min at a spin pump speed of 2 rpm). Furthermore, the spinneret features 600 nozzles spaced at 8 mm intervals, so interference caused by interaction between the individual filaments cannot be excluded. The collector plate of the pilot-scale device has an area of 467 cm^2^, which equates to only 0.79 cm^2^ for each nozzle, so individual filaments may restrict and suppress each other during whipping movements. At the pilot scale, switching from positive to negative polarity reduced the fiber diameter with a similar trend to that observed at the laboratory scale. However, one of the major challenges observed during scale-up was the non-sticking nature of the filaments. The fibers produced by the pilot-scale device did not resemble non-woven-like structures but rather 6oo individual, unconnected filaments, probably because the PBS filaments solidified immediately after leaving the nozzle before reaching the collector. The throughput at the pilot-scale machine (0.001 g/min at 2 rpm, 0.003 g/min at 5 rpm, 0.007 g/min at 10 rpm) is even at the highest spin pump speed of 10 rpm, 10 times smaller than the throughput of the laboratory-scale device (0.081 g/min).

The significantly lower throughput leads to a much slower fiber deposition during pilot-scale melt electrospinning, which gives sufficient time for crystallization of PBS fibers after leaving the nozzle and before reaching the collector. This leads to the formation of fibers that do not stick to each other. Furthermore, this fast crystallization can also hinder the interaction with the electric field. Therefore, an approach to keep the fibers warm after extruding in order to delay the crystallization, increasing the time for whipping behavior, which is important for stretching of fibers to small diameters, remains necessary. This could also lead to filaments sticking to each other during the fiber deposition. This could be addressed by adding an ~80 °C heating step to stick the filaments together or by enclosing the spinneret in a climate chamber to delay the solidification of the polymer, as previously shown during the pilot-scale melt electrospinning of PLA [14].

A nozzle-to-collector distance of 6 cm was sufficient to induce whipping motions and Taylor cone formation at the laboratory scale, but at the pilot scale, we observed sparks jumping from the tip of the nozzle, indicating it was too close to the collector. Increasing the distance to 7 cm eliminated this issue and fiber deposition was possible. Our experiments have shown that the transferability of melt electrospinning from the laboratory to the pilot-scale device is only possible to a limited extent and requires careful evaluation.

## 4. Conclusions

We determined the effect of collector plate polarity on the melt-electrospinning of PBS at the laboratory scale as a strategy to minimize the final fiber diameter. We also investigated the transferability of results from a laboratory-scale single-nozzle device to our 600-nozzle pilot-scale prototype. In both cases, applying a negative voltage to the collector significantly reduced the overall fiber diameter. The finest fibers produced in the laboratory had an average diameter of 32.26 µm, whereas the finest fibers produced using the pilot-scale device had an average diameter of 10.88 µm. The fiber webs produced in the laboratory and deposited on the negatively charged collector were arranged more densely and uniformly than those deposited on the positively charged collector, and they covered a larger surface area. Similar results were observed at the pilot scale, although the fibers did not stick together and resembled a loose web rather than a non-woven-like structure.

We believe the reason for that might be based on the differences in throughput. The significantly lower throughput leads to a much slower fiber deposition on the collector at pilot-scale than at lab-scale, leading to crystallization of PBS before reaching the collector. The fast solidification gives PBS less time to interact with the electric field. Therefore, the usage of a climate chamber might be investigated in future in order to delay the solidification of PBS fibers at pilot-scale in order to give the extruded fibers more time for whipping behavior to occur and by this resulting in fibers even lower diameters. Additionally, by integrating a climate chamber around the spinneret the solidification of the fibers might be delayed to that extent that the fibers solidify after reaching the collector and thus producing a non-woven-like structure of filaments that stick. In addition, the pilot-scale melt-electrospinning process might need to be optimized for each throughput and the appropriate nozzle-to-collector distance will need to be investigated. Thereby, even higher electric field strength might need to be considered to further reduce the fiber diameters in future.

This result highlights the practical differences between laboratory-scale and pilot-scale melt-electrospinning despite them being based on the same principles. This limits the transferability of laboratory results to the pilot scale and suggests that the solidification of PBS fibers must be delayed during pilot-scale melt-electrospinning. Nevertheless, changing the polarity of the collector is a promising and inexpensive modification that can reduce the diameter of PBS fibers. Our results can therefore be used to tailor the melt-electrospinning of biobased polymers to produce fibers with specific diameters.

## Figures and Tables

**Figure 1 polymers-14-02865-f001:**
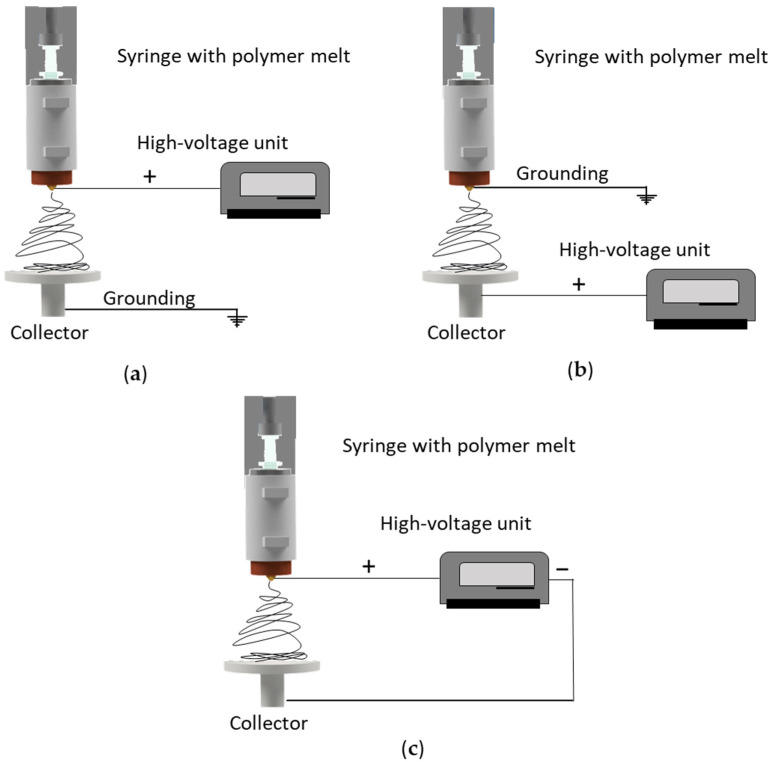
Typical current electrospinning setups. (**a**) Syringe charged and collector grounded. (**b**) Syringe grounded and collector charged. (**c**) Both syringe and collector charged.

**Figure 2 polymers-14-02865-f002:**
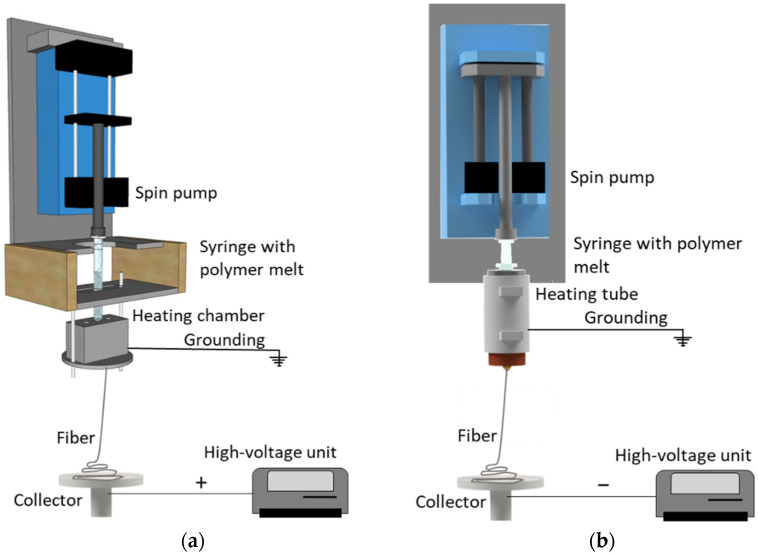
Our single-nozzle laboratory-scale melt-electrospinning equipment. (**a**) Original setup before optimization, including a heating chamber consisting of several individual elements. (**b**) Modified setup including a continuous heating tube tailored to the syringe.

**Figure 3 polymers-14-02865-f003:**
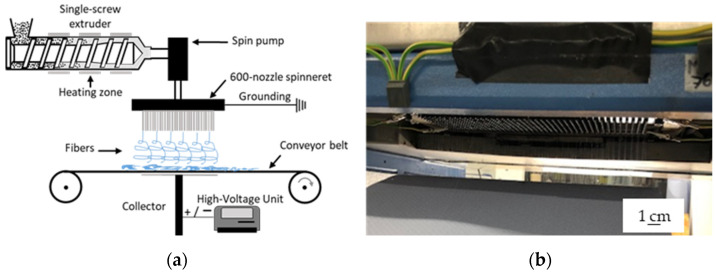
Our pilot-scale melt-electrospinning prototype. (**a**) Setup of the pilot-scale electrospinning machine. (**b**) Grounded spinneret including 600 nozzles.

**Figure 4 polymers-14-02865-f004:**
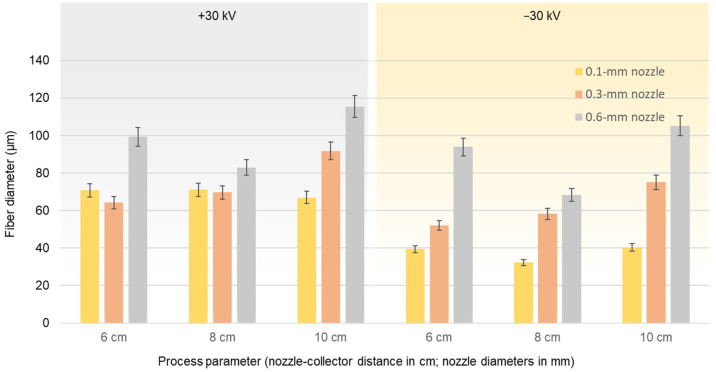
Diameter of PBS fibers produced with a throughput of 0.1 mL/min at a polymer melt temperature of 235 °C with three different nozzle-to-collector distances and nozzles of different diameters as indicated by the color key. We directly compared electric field strengths of +30 kV and −30 kV. Data are presented as means ± standard deviation (*n* = 100).

**Figure 5 polymers-14-02865-f005:**
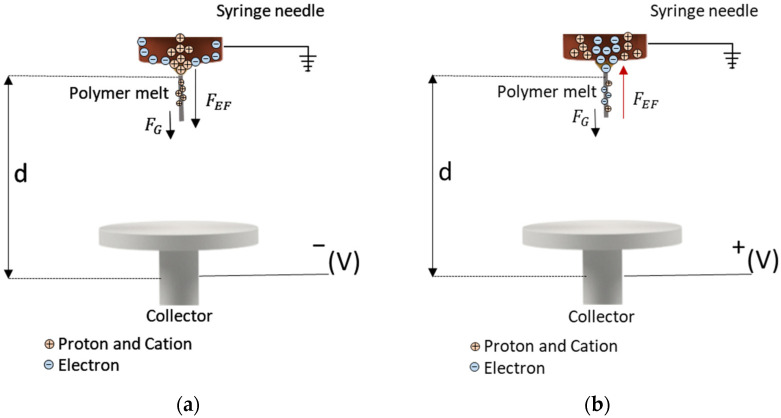
Motion of charges with a grounded syringe needle tip and (**a**) a negatively charged collector or (**b**) a positively charged collector. F*_G_* = gravitational force. F*_EF_* = electrical field force.

**Figure 6 polymers-14-02865-f006:**
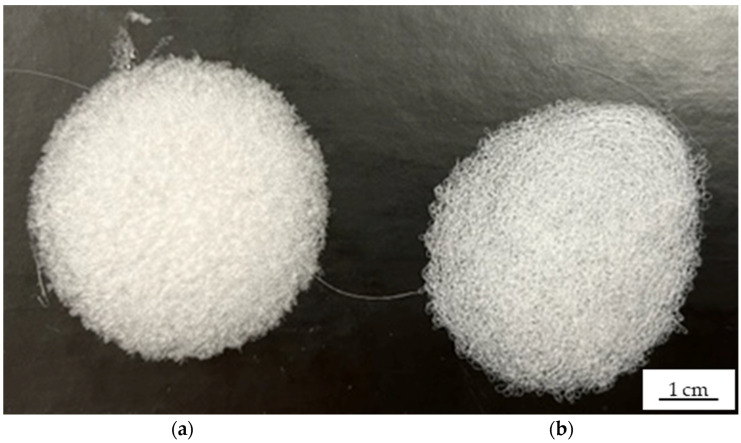
Fiber web samples produced using single-nozzle laboratory-scale melt-electrospinning equipment. (**a**) Negatively charged (−30 kV) collector. (**b**) Positively charged (+30 kV) collector.

**Figure 7 polymers-14-02865-f007:**
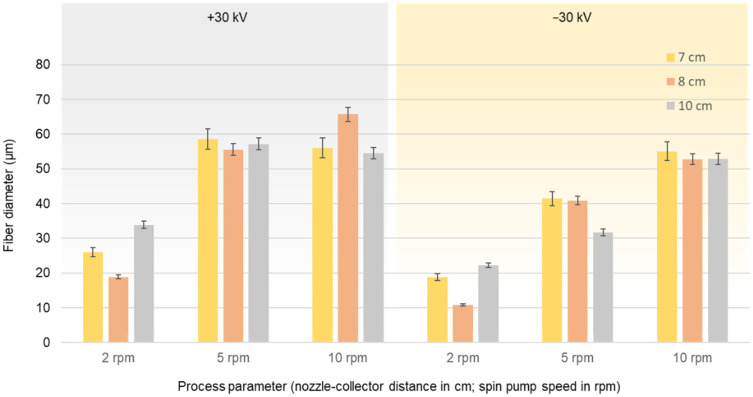
Diameter of PBS fibers produced using pilot-scale melt-electrospinning equipment at a spinneret temperature of 235 °C, a spin pump speed of 2, 5, or 10 rpm, a nozzle-to collector distance of 7, 8, or 10 cm, 600 nozzles (diameter 0.6 mm each), and an electric field strength of +30 or −30 kV. Data are means ± standard deviation (*n* = 100).

**Figure 8 polymers-14-02865-f008:**
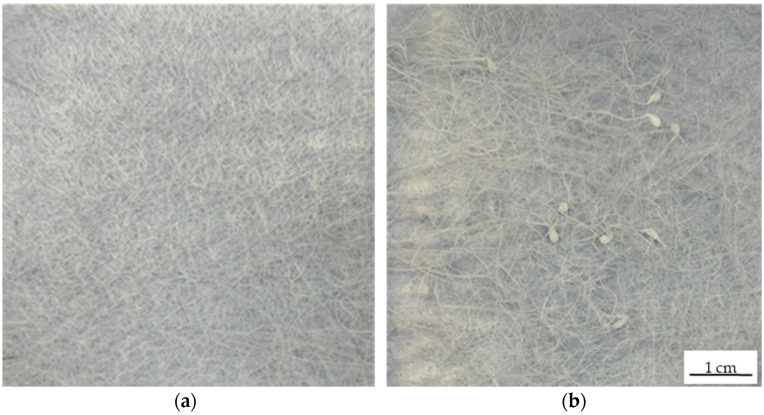
Fiber web samples produced using the pilot-scale melt-electrospinning equipment at a pump speed of 5 rpm and a nozzle-to-collector distance of 10 cm. (**a**) Negatively charged (−30 kV) collector. (**b**) Positively charged (+30 kV) collector.

**Figure 9 polymers-14-02865-f009:**
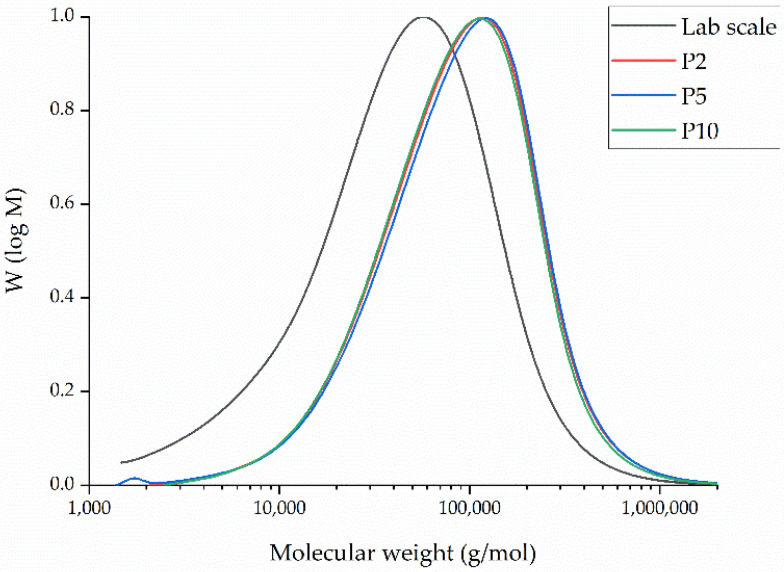
Molar mass distribution curves of PBS fibers from laboratory-scale (throughput of 0.1 mL/min) and pilot-scale 2 (P2), 5 (P5) and 10 (P10) rpm melt-electrospinning, both at 235 °C.

**Table 1 polymers-14-02865-t001:** Process parameters for the laboratory-scale melt-electrospinning experiments.

Trial	Nozzle Diameter (mm)	Nozzle–Collector Distance (cm)	Electric Field (kV)
1	0.10	6	+30
2	0.10	8	+30
3	0.10	10	+30
4	0.10	6	−30
5	0.10	8	−30
6	0.10	10	−30
7	0.30	6	+30
8	0.30	8	+30
9	0.30	10	+30
10	0.30	6	−30
11	0.30	8	−30
12	0.30	10	−30
13	0.60	6	+30
14	0.60	8	+30
15	0.60	10	+30
16	0.60	6	−30
17	0.60	8	−30
18	0.60	10	−30

**Table 2 polymers-14-02865-t002:** Process parameters for the pilot-scale melt-electrospinning experiments.

Trial	Spin Pump Speed (rpm)	Nozzle–Collector Distance (cm)	Electric Field (kV)
1	2	7	+30
2	2	7	+30
3	2	7	+30
4	2	7	−30
5	2	7	−30
6	2	7	−30
7	5	8	+30
8	5	8	+30
9	5	8	+30
10	5	8	−30
11	5	8	−30
12	5	8	−30
13	10	10	+30
14	10	10	+30
15	10	10	+30
16	10	10	−30
17	10	10	−30
18	10	10	−30

**Table 3 polymers-14-02865-t003:** The weight average relative molecular weight (Mw), number average molar mass (Mn), and polydispersity index (PDI) of PBS fibers processed at laboratory-scale (throughput of 0.081 g/min) and pilot-scale (throughput of 0.001, 0.003, and 0.007 g/min), both at 235 °C.

Melt-Electrospinning Process	Throughput (g/min)	Mn (Da)	Mw (Da)	PDI
Lab Scale	0.081	22.560	99.440	4.41
Pilot Scale (2 rpm)	0.001	52.720	132.300	2.51
Pilot Scale (5 rpm)	0.003	49.690	134.100	2.69
Pilot Scale (10 rpm)	0.007	52.060	126.300	2.43

## Data Availability

The datasets used and/or analyzed during this study are available from the corresponding author on reasonable request.

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
