# Peer review of "The Effect of Electrical Polarity on the Diameter of Biobased Polybutylene Succinate Fibers during Melt Electrospinning"

_polymers, 2022, doi:10.3390/polym14142865_

Round 1

Reviewer 1 Report

The article from Maike-Elisa Ostheller, et al. involves evaluating the electrical polarity effect on the electrospun fiber diameter of polybutylene succinate in a lab-scale process and a pilot-scale process. I believe the topic is interesting and informative for the readers of Polymers and the article was well written. However, I would like the authors to address some issues in the manuscript before it’s considered for publication.

-Lack of discussing relevant process variables. For example, melt throughput is an important parameter of the spinning process. However, there is no explanation on why 0.1 mL/min of polymer melt was chosen for the lab-scale process and how it compared to the throughputs used in pilot-scale process. Also, ambient temperature and humidity are relevant to the rate of solidification of the polymer melt and hence affect the physical properties of the fibers. Particularly, they could contribute the observed product property differences between the lab-scale process and the pilot-scale process. However, they were not measured and mentioned.

-Lack of adequate product characterizations. While I understand the authors would like to focus on the diameter of the fibers, I believe providing other relevant characterization results will make this article more valuable to its readers. For example, GPC results might help understanding the impact of the dwell time difference between the lab-scale process and the pilot-scale process; and DSC results should provide some insights in the relationship between different process settings and the product physical properties.

Author Response

Thank you very much for your valuable input. We have taken your comments into consideration and have made changes accordingly and hope that the article now meets your expectations. 

Reviewer 2 Report

Electrospinning is a simple and cost-effective process for the manufacturing of microfibers and nanofibers, with many advantages over alternative techniques such as drawing, template synthesis, and phase separation. Conventional fiber-spinning processes such as melt spinning and wet spinning draw and stretch fibers by applying a mechanical force, whereas electrospinning uses an electric field to produce continuous fibers. Melt electrospinning is a simple, versatile and widely-used technique for the production of microfibers and sub-microfibers. Polybutylene succinate (PBS) is a promising raw material for the preparation of melt-electrospun fibers at the laboratory-scale. In this manuscript, the authors determined the effect of collector plate polarity on the melt electrospinning of PBS at the laboratory scale as a strategy to minimize the final fiber diameter. the authors also investigated the transferability of results from a laboratory-scale single-nozzle device to our 600-nozzle pilot-scale prototype. The topic is important, the results are interesting and the methodology followed is appropriate, while the content falls well within the scope of this Journal. In general the paper makes fair impression and my recommendation is that it merits publication in this Journal, after the following major revision:

1. The detailed literature review indicates efforts made by the authors. The coherence of the related work, however, is still not clear. It may help the authors by answering the following questions: Why are these works relevant? Which specific problems were addressed? How are the previous results related with the latest work? What are the outstanding, unresolved, research issues? Which of them has been solved by the proposed study? Answering the questions leads to the novelty of the proposed work naturally. I think this is essential to keep the interest of the reader.

2. Materials and methods part, although the results look “making sense”, the current form reads like a simple lab report. The authors should dig deeper in the results by presenting some in-depth discussion.

3. The fiber has been widely used in the industry. The fiber is found to serve in many practical applications, such as porous materials, (see [Powder Technology, 2019, 349:92-98; International Journal of Heat and Mass Transfer, 2019, 137:365-371). Authors should introduce some related knowledge to readers. I think this is essential to keep the interest of the reader.

4. The finest fibers produced in the laboratory had an average diameter of 32.26 µm, whereas the finest fibers produced using the pilot-scale device had an average diameter of 10.88 µm. The fiber webs produced in the laboratory and deposited on the negatively charged collector were arranged more densely and uniformly than those deposited on the positively charged collector, and they covered a larger surface area. Similar results were observed at the pilot scale, although the fibers did not stick together and resembled a loose web rather than a nonwoven-like structure. This result highlights the practical differences between laboratory-scale and pilot-scale melt electrospinning despite them being based on the same principles. The authors should give some explanation on above conclusions and data.

5. Please expand the motivation, the problem context, clarify the problem description, and (if possible) add specific objectives.

6. Please, expand the conclusions in relation to the specific goals and the future work.

Author Response

(The authors gave the same response as above.)

Round 2

Reviewer 1 Report

The major issues in the original manuscript have been addressed.

Author Response

Thank you again for the valuable comments and critical remarks that have enhanced our manuscript.

Reviewer 2 Report

It is ok.

Author Response

(The authors gave the same response as above.)
